# microRNA-185 Inhibits SARS-CoV-2 Infection through the Modulation of the Host’s Lipid Microenvironment

**DOI:** 10.3390/v15091921

**Published:** 2023-09-14

**Authors:** Nadine Ahmed, Magen E. Francis, Noreen Ahmed, Alyson A. Kelvin, John Paul Pezacki

**Affiliations:** 1Department of Chemistry and Biomolecular Sciences, University of Ottawa, Ottawa, ON K1N 6N5, Canada; 2Department of Biochemistry, Microbiology, and Immunology, University of Saskatchewan, Saskatoon, SK S7N 5A2, Canada; 3Vaccine and Infectious Disease Organization-International Vaccine Centre (VIDO-InterVac), University of Saskatchewan, Saskatoon, SK S7N 5E3, Canada

**Keywords:** SARS-CoV-2, Metabolism, microRNA, SREBP, antiviral mechanisms

## Abstract

With the emergence of the novel *betacoronavirus* Severe Acute Respiratory Syndrome Coronavirus 2 (SARS-CoV-2), there has been an urgent need for the development of fast-acting antivirals, particularly in dealing with different variants of concern (VOC). SARS-CoV-2, like other RNA viruses, depends on host cell machinery to propagate and misregulate metabolic pathways to its advantage. Herein, we discovered that the immunometabolic microRNA-185 (miR-185) restricts SARS-CoV-2 propagation by affecting its entry and infectivity. The antiviral effects of miR-185 were studied in SARS-CoV-2 Spike protein pseudotyped virus, surrogate virus (HCoV-229E), as well as live SARS-CoV-2 virus in Huh7, A549, and Calu-3 cells. In each model, we consistently observed microRNA-induced reduction in lipid metabolism pathways-associated genes including SREBP2, SQLE, PPARG, AGPAT3, and SCARB1. Interestingly, we also observed changes in angiotensin-converting enzyme 2 (ACE2) levels, the entry receptor for SARS-CoV-2. Taken together, these data show that miR-185 significantly restricts host metabolic and other pathways that appear to be essential to SAR-CoV-2 replication and propagation. Overall, this study highlights an important link between non-coding RNAs, immunometabolic pathways, and viral infection. miR-185 mimics alone or in combination with other antiviral therapeutics represent possible future fast-acting antiviral strategies that are likely to be broadly antiviral against multiple variants as well as different virus types of potential pandemics.

## 1. Introduction

Severe Acute Respiratory Syndrome Coronavirus 2 (SARS-CoV-2) is the etiological agent which results in the development of COVID-19 [1]. SARS-CoV-2 has been a global concern over the past few years and therefore international efforts were put in place to rapidly curb viral spread and limit complications associated with the infection [2]. This disease of concern is characterized by several severe clinical outcomes, which include acute respiratory distress syndrome, septic shock, and multiple organ dysfunction syndromes [3]. Thus, unprecedented efforts were established for rapid drug discovery and the development of potent antiviral therapeutic strategies [4]. Some of these strategies include neutralizing antibodies, vaccines, and therapeutic antiviral drugs [4,5,6,7]. Fortunately, effective vaccine development and production rapidly evolved over the past year, aiding in curbing the spread of the virus globally [2,8]. However, with the emergence of other SARS-CoV-2 variants of concern (VOCs), which are more resistant to the neutralization effects of approved vaccines [9,10,11], there is an urgent need for the development and the identification of potent therapeutic antivirals to aid in the suppression of the severe consequences of the infection. Hence, improved understanding of SARS-CoV-2 host–virus interactions would aid in delineating molecular pathways and mechanisms essential for the SARS-CoV-2 life cycle.

Both SARS-CoV and SARS-CoV-2 share similar cellular tropisms—preferentially infecting the lower respiratory tract and utilizing angiotensin-converting enzyme 2 (ACE2) as an entry receptor [12]. It has been found that the ACE2 receptor is localized in lipid rafts [12], which may indicate the involvement of cholesterol and glycosphingolipids in modulating SARS-CoV-2 and ACE2 interactions since these lipid components are enriched in these lipid microdomains [13,14]. SARS-CoV-2 also has been shown to be highly dependent on host-cell metabolic pathways that provide components and energy needed for the creation of progeny viruses [15].

Recently, microRNAs (miRNAs) have emerged as essential regulators of these pathways and even have direct antiviral activity [16]. We have previously shown that miR-185 inhibits hepatitis C and dengue viruses through alterations in the host’s lipid microenvironment [17]. In this study, we focus on delineating the effects of miR-185 on the modulation of SARS-CoV-2 entry and replication. Initially, we investigated the role of miR-185 in modulating the entry of SARS-CoV-2 using a Spike protein pseudotyped virus model. To further evaluate the impact of this miRNA on the SARS-CoV-2 VOC, we utilized the same model to assess the impact of the miRNA on Spike-harbouring mutations of interest such as D614G and N501Y and the Alpha, Beta, Delta, and Omicron VOCs. Next, we investigated the role of the miRNA in low-pathogenicity coronavirus 229E infection (HCoV-229E). Finally, we validated our findings with the replication-competent SARS-CoV-2 cell culture model. Our study highlights the impact of miR-185 in antagonizing SARS-CoV-2 infection and underscores the therapeutic potential of miRNAs in decreasing the burden of SARS-CoV-2 and COVID-19.

## 2. Results 

### 2.1. Expression of SARS-CoV-2 Spike Protein and Pseudovirus Incorporation of SARS-CoV-2 Spike (S) Protein

To examine the effects of miR-185 overexpression on viral entry, we sought to utilize the lentiviral pseudovirus system expressing the SARS-CoV-2 Spike (S) protein. To ensure optimal production of our engineered particles, a plasmid-expressing SARS-CoV-2 S protein lacking the terminal 19 amino acids containing the endoplasmic reticulum retention signal was utilized [18]. The HIV-based retroviral system was used to produce the engineered pseudovirus (Figure 1A). The HIV-based system consisting of HIV-1 NL4-3 ΔEnv Vpr Luciferase Reporter Vector (pNL4-3.Luc.R-E-) conveniently expresses a Luciferase gene that allows for the detection of viral entry upon infection [19]. Initially, to confirm the successful production of the engineered particles, we validated the expression of the SARS-CoV-2 S protein in Hek293T producer cells and the incorporation of the protein into pseudovirus using Western blotting with an anti-Spike antibody predicted to recognize the S2 subunit. As expected, the S protein was readily detected using the anti-Spike antibody in cells transfected with the S encoding vector and engineered pseudovirus (Figure 1B). We additionally assessed the functionality of the produced pseudotyped virus by comparing the Luciferase activity of cells following the infection with Spike protein pseudotyped virus relative to a mock virus lacking Spike expression. The mock virus resulted in minimal background Luciferase activity in recipient cells relative to the activity observed following infection with a pseudotyped virus (Appendix A). These data confirm the functionality and the competency of the produced virus. Consequently, this system is utilized in our study to report on Spike-based entry of SARS-CoV-2.

### 2.2. miR-185 Antagonizes SARS-CoV-2 S Protein Pseudotyped Virus Entry in Huh7 and Calu-3 Cell Lines

In order to assess the impact of miR-185 overexpression on the entry of SARS-CoV-2 pseudovirions incorporating SARS-CoV-2 S protein, we transfected miR-185 or control miRNA (con-miR) mimics or inhibitors in Calu-3 and Huh7 cells, which were previously validated to endogenously express sufficient levels of ACE2 to mediate SARS-CoV-2 entry [20]. Following transfection, the cells were infected with equal amounts of pseudovirus at MOI of 1 for 48 h. Cells were lysed, and a Luciferase assay was performed to examine the levels of Luciferase signal in con-miR and miR-185-transfected cells. Interestingly, miR-185-overexpressing cells displayed a significant decrease in Luciferase signal relative to con-miR transfected cells, suggesting a decrease in pseudovirus entry in both cell lines (Figure 2A,B and Appendix A). Additionally, we sought to investigate the effects of inhibition of the endogenous levels of miR-185 on viral entry. Interestingly, in miR-185-inhibitor-transfected Huh7 cells, an increase in viral entry was observed (Figure 2B and Appendix A). Overall, these findings suggest that miR-185 alters endogenous mechanisms essential for viral entry to antagonize SARS-CoV-2 Spike-based entry into these cells.

Accumulation of mutations in the SARS-CoV-2 Spike protein has enhanced binding to its receptor and led to the emergence of VOCs displaying higher disease severity, resistance to neutralizing antibodies elicited by current vaccines or from previous infection, and and higher resistance to treatment [21,22]. Thus, we sought to investigate the effects of miR-185 mimic transfection on the entry of virions engineered to express S proteins harbouring some of the Spike mutations known to enhance Spike-based entry into host cells.

The D614G and N501Y mutations are found to be responsible for the increased transmissibility and fitness of the B.1.1.7 and the B.1.351 variants [21,22]; thus, entry assays were performed with the pseudovirus harbouring these mutations. Western blot analysis validated the successful expression of these mutants on pseudotyped particles (Figure 2C). Interestingly, miR-185 is found to suppress the entry of the D614G and N501Y mutants in both Calu-3 and A549 cells expressing ACE2 (Figure 2D–G and Appendix A–F). Finally, we aimed to evaluate whether miR-185’s inhibitory effects are extended to some of the current VOCs. To examine the effects of miR-185 on the cellular entry of these VOCs, we similarly produced and validated the expression of Alpha-, Beta-, Delta-, and Omicron-variant S proteins on the pseudotyped virus (Figure 3A). Interestingly, miR-185 overexpression ubiquitously suppresses SARS-CoV-2 entry despite variations in S sequence in Calu-3 (Figure 3B–E and Appendix A–D) and Huh7 cells (Appendix A–C). Overall, these findings suggest that miR-185 may play a role in SARS-CoV-2 pathogenesis, potentially by altering the cellular microenvironment to impede binding and entry.

### 2.3. Inhibition of SREBP2-Modulated Signaling Antagonizes SARS-CoV-2 Spike Pseudotyped Viral Entry

We have previously shown that miR-185 expression is induced by 25-hydroxycholesterol (25-HC) [17], which is an oxysterol that displays broad antiviral properties against various families of enveloped viruses, which includes coronaviruses [23,24,25,26]. This lipid effector synthesis from cholesterol is catalyzed by cholesterol-25-hydroxylase, whose expression is induced by interferon (IFN) in macrophages as an innate response to viral infection [27]. Interestingly, 25-HC has also been shown to have antagonizing effects on viral-cell fusion and viral entry [28,29]. Furthermore, 25-HC has been found to inhibit cholesterol synthesis by targeting the activity of sterol regulatory element-binding proteins (SREBPs), which are membrane-bound transcription factors that modulate the synthesis and expression of genes involved in lipid and cholesterol biosynthesis [30]. 

Thus, we sought to validate the effects of cholesterol depletion on viral entry using our model by treating Huh7 cells with cholesterol-depleting small molecules such as 25-HC and Fluvastatin, an HMG-CoA reductase inhibitor that suppresses cholesterol synthesis. We treated cells with varying concentrations of both 25-HC and Fluvastatin and we found a dose-dependent decrease in the levels of pseudovirus entry, with the most potent inhibition at 5 µM for 25-HC and 10 µM for Fluvastatin (Appendix A, respectively). These findings suggest that dysregulation of cholesterol levels mediated by 25-HC and Fluvastatin treatments antagonizes SARS-CoV-2 S pseudovirus entry and overall suggest that the antiviral effects mediated by these compounds likely affect early stages of the viral life cycle.

Membrane cholesterol and lipid composition have been found to be indispensable for enveloped viral entry and attachment. It was shown that ACE2 is localized to cholesterol-rich lipid rafts, suggesting that cholesterol may play an essential role in modulating several steps in the SARS-CoV-2 life cycle since cholesterol was found to play extensive roles in the life cycles of other pathogens including SARS-CoV [13,31,32]. Additionally, cholesterol depletion has been found to disrupt the virion membrane and binding affinity to the host cell, while cholesterol-rich regions in the host’s membrane have been found to be instrumental in Spike-mediated fusion and viral entry [33]. Moreover, a decrease in the levels of membrane-bound cholesterol in host cells results in a reduction in Spike binding to ACE2 and a decrease in infectivity [34]. Our data with 25-HC and Fluvastatin support an important role for de novo cholesterol biosynthesis in the SARS-CoV-2 life cycle, which is particularly important for cell entry mechanisms.

### 2.4. miR-185 Inhibits SARS-CoV-2 Entry by Modulating Lipid Metabolism and Repression of ACE2 Expression in Calu-3 Cells

Given that miR-185 is a 25-HC-regulated miRNA, we sought to evaluate the mechanism by which this miRNA inhibits viral entry. We have previously shown that miR-185 is an immunometabolic miRNA that strongly regulates lipid and cellular metabolism and thus alters the lipid microenvironment [17]. Specifically, overexpression of miR-185 in Huh7 cells was shown to decrease lipid droplet formation, cellular triglyceride levels, and cholesterol biosynthesis [17]. Given this, we sought to evaluate the levels of genes involved in these processes following miR-185 overexpression. miR-185 directly targets SREBP2, a transcription factor that modulates the expression of cholesterol biosynthetic genes and is a master regulator of these processes [35]. 

We first aimed to confirm that miRNA transfection does not affect cell viability. An MTT assay was conducted and confirmed minimal to no effects on cell viability following transfection of the mimics (Appendix A). Following this confirmation, we aimed to evaluate the levels of SREBP2 mRNA using qPCR following miR-185 transfection in Calu-3 cells. A significant decrease in the levels of SREBP2 was observed in cells overexpressing miR-185 relative to con-miR-transfected cells. This suggests that miR-185 likely alters cholesterol metabolism by targeting and suppressing a master regulator of this process. 

To further confirm miR-185’s inhibitory effects on cholesterol biosynthesis, we evaluated the levels of squalene epoxidase (SQLE) which encodes an enzyme that catalyzes cholesterol synthesis and whose expression is regulated by SREBP2. SQLE was significantly downregulated in cells overexpressing miR-185 (Figure 4A). We have further confirmed the decrease in the protein levels of SREBP2 and SQLE in miR-185-overexpressing Calu-3 cells relative to con-miR-transfected cells (Figure 4B). Additionally, we show that miR-185 suppresses the levels of another lipogenic direct target, HDL-scavenger receptor B type 1 (SCARB1). This protein is known to facilitate the ACE2-dependent entry of SARS-CoV2 [36], contributing to miR-185’s suppressive effects on entry. In fact, treatment of cultured cells with pharmacological SCARB1 inhibitors inhibits SARS-CoV-2 infection [36]. Overall, these findings suggest that miR-185’s inhibitory effects are modulated by the direct or indirect repression of cholesterol-metabolism-associated genes, such as SREBP2, SQLE, and SCARB1.

In addition to repressing genes involved in cholesterol biosynthesis and metabolism, miR-185 suppresses the levels of other target genes and lipogenic transcription factors involved in fatty acid and triglyceride biosynthesis and regulation; for example, we show that AGPAT3, an enzyme that catalyzes steps in the biosynthesis of triglycerides from glycerol-3-phosphate (G3P) [37], is significantly downregulated in miR-185-overexpressing cells relative to con-miR transfected cells (Figure 4A). Although ACE2 is not a predicted target of miR-185, we observed a decrease in the levels of ACE2 mRNA in miR-185-overexpressing cells (Figure 4C). Interestingly, the PPARγ lipogenic transcription factor was previously found to regulate ACE2 expression [38]. In Calu-3 cells overexpressing miR-185, PPAR-gamma expression is downregulated (Figure 4A), and we hypothesize dysregulations in the levels of PPAR-γ may result in the observed decrease in the levels of ACE2 and ultimately contribute to miR-185’s ability to suppress SARS-CoV-2 entry into host cells. Overall, these findings confirm miR-185’s function in modulating lipid metabolic processes in Calu-3 cells and point towards a mechanism by which miR-185 potentially regulates the anti-SARS-CoV-2 response observed in cells overexpressing the miRNA.

### 2.5. miR-185 Inhibits HCoV-229E Replication and Infectivity

Next, we sought to evaluate the effects of miR-185 overexpression on the replication and the life cycle of live coronavirus infection. We first evaluated the effects of a low-pathogenic human coronavirus 229E (HCoV-229). Initially, we assessed the effects on viral replication by quantifying the levels of intracellular viral RNA levels in Huh7 cells overexpressing miR-185. Huh7 cells were used in these experiments since they are permissive to HCOV-229E viral infection. Interestingly, a significant decrease in the levels of the intracellular virus was observed (Figure 5A). Consistent with that, a decrease in infectivity was observed (Figure 5B). A decrease in viral replication was also reflected in the A549 lung carcinoma cell line, which confirms that miR-185’s suppressive effect on HCoV is observed in cells that represent a primary site of infection (Appendix A). 

Next, we pursued to examine miR-185’s effect on cholesterol biosynthesis in Huh7 cells during HCoV-229E infection. Consistent with previous findings observed in Calu-3 cells, a decrease in SREBP2 levels was observed along with a decrease in SREBP2 target, SQLE (Figure 5C). Additionally, a decrease in metabolic targets of miR-185 was observed, with a significant decrease in the levels of AGPAT3 and SCARB1 (Figure 5C). Overall, these findings validate the suppressive antiviral effects of miR-185 in an infectious model, and these effects are likely modulated through miR-185’s inhibitory effects on metabolic pathways that are found to enhance viral entry, replication, and infectivity. 

### 2.6. miR-185 Overexpression Inhibits SARS-CoV-2 Pathogenesis in Calu-3 Cells 

Finally, to investigate whether miR-185 could modulate SARS-CoV-2 infection, we transfected Calu-3 cells with con-miR or miR-185 mimics and 24 h post-transfection we infected the cells with SARS-CoV-2 at an MOI of 0.01. Intracellular and extracellular viral levels were investigated at 24, 48, and hours post-infection. Over two log_10_ fold changes (TCID50/mL) were observed in intracellular and extracellular virus levels at all investigated time points (Figure 6A,B), further confirming the inhibitory effects of miR-185 on coronavirus entry and infection *in vitro*. The suppression of SREBP2 and AGPAT3, which are predicted and validated direct targets of miR-185, was confirmed to be downregulated in miR-185-overexpressing cells in all tested time points (Appendix A). These findings further validate that miR-185 attenuates SARS-CoV-2 viral infection by inhibiting lipid and sterol metabolic pathways which promote coronavirus infection in human cells.

## 3. Discussion 

microRNAs are non-coding RNAs that have been found to extensively regulate diverse aspects of cellular pathways and mechanisms [39,40,41]. They are considered to be master regulators of gene expression and they can modulate diverse pathways; thus, it is not surprising to observe dysregulated miRNA profiles in altered cellular states as a result of disease or infection [41,42,43]. Several miRNAs have emerged of critical importance in the modulation of viral infections and, therefore, miRNAs may represent a target for therapeutic investigation [17,43,44,45]. For example, miR-122 is a miRNA that has been found to play an integral role in promoting HCV replication, and therefore miR-122 represents a potential therapeutic target for attenuating HCV pathogenesis. In fact, miravirsen and RG-101, experimental anti-miR-122 oligonucleotides, were developed for clinical use and were found to decrease HCV RNA levels in chronic HCV patients through phase 1 and phase 2 trials [46,47]. Therefore, it is of critical importance to evaluate the role of miRNAs in the modulation of virus infection since they may represent pro or antiviral host factors that may be utilized to control viral pathogenesis.

In this study, we evaluate the role of miR-185 in the modulation of SARS-CoV2 pathogenesis and examine the potential mechanism by which the miRNA acts to inhibit viral entry, replication, and infectivity. We have previously shown that miR-185’s expression is regulated by 25-HC, an antiviral molecule produced by cholesterol 25-hydroxylase, an enzyme that catalyzes the oxidation of cholesterol to 25-HC [17,48,49]. Broad antiviral properties of this molecule have been demonstrated with recent reports highlighting the potency of 25-HC in inhibiting SARS-CoV-2 infection *in vitro* and *in vivo* [50]. 

In this study, we initially generated SARS-CoV-2 Spike-pseudotyped viral particles to examine the effects of miR-185 on viral entry. The entry mechanism of SARS-CoV-2 has been extensively studied. Generally, on mature viruses, the Spike protein responsible for binding to the cellular receptor is present in a trimeric form, containing an S1 head and an S2 stalk. S1 contains the receptor binding domain (RBD) and recognizes the cellular receptor ACE2 [12] upon recognition, a subsequent viral entry where viral and lysosomal membrane fusion eventually takes place. In addition to ACE2, other host cellular proteases such as TMPRSS2 have been identified to aid in the viral entry process [12]. Interestingly, cholesterol and other lipid species significantly affect the SARS-CoV-2 entry process. As a modulator of lipid metabolism [17], we sought to investigate the effects of miR-185 on SARS-CoV-2 entry and pathogenesis.

We show that miR-185 suppresses entry of the pseudotyped-virus-expressing WT Spike, D614G, and N501Y mutants, and the Alpha, Beta, Delta, and Omicron Spike. This has led us to believe that miR-185 dysregulates pathways that are essential for the entry of SARS-CoV-2 despite the nature of the Spike protein and thus represents a pan-antiviral mechanism against current and potential SARS-CoV-2 variants. We show that miR-185 dysregulates the entry of viral particles by modulating lipid metabolic pathways, which include pathways involved in cholesterol and triglyceride biosynthesis. This was evidenced by miR-185’s significant suppressive effects on the expression of genes regulating these processes. Specifically, we show miR-185 targets and decreases the levels of endogenous SREBP2, a transcriptional regulator of cholesterol biosynthesis, and its targets, thereby dysregulating the levels of cholesterol in the cells and influencing SARS-CoV-2 entry and infectivity.

Lipids’ involvement in SARS-CoV-2 pathogenesis is now being extensively investigated. Several studies have revealed that during human coronavirus infection the cell’s lipid profile is significantly dysregulated [34,51]. A recent study showed that similar to other (+)RNA viruses, SARS-CoV-2 induces lipid droplet formation, a phenotype not observed during SARS-CoV infection [52]. These lipid droplets are usually exploited by RNA viruses to obtain energy to support their replication or acquire lipids for particle formation. This phenotype was consistently observed in vivo and *in vitro* during SARS-CoV-2 infection, similar to observations in hepatocytes following HCV infection [53]. Additionally, it has been shown that cholesterol localized in the lipid rafts is an essential entry factor for coronaviruses, both *in vitro* and *in vivo* [13,34], and a determinant of SARS-CoV-2 pathogenesis and replication. Sterols and oxysterols significantly influence the mechanisms of viral infections and cellular response to infections [54]. Lipid rafts are indispensable for the entry and internalization of coronaviruses, where cholesterol depletion from cellular membranes was found to decrease viral entry [55,56,57]. Specifically for SARS-CoV-2, it was shown that localization of cholesterol in lipid rafts is crucial for infectivity and entry which suggests that metabolic remodeling in the host cell is an essential factor and a determinant of viral pathogenesis [57]. Thus, our study shows that a miR-185-mediated decrease in the levels of endogenous cholesterol contributes to a decrease in viral entry and infectivity. Furthermore, we show that miR-185 overexpression significantly suppresses HCoV-229 and SARS-CoV-2 replication and infectivity. 

Our study highlights the therapeutic potential of miRNAs in regulating virus infections. Generally, miRNAs with antiviral properties can be utilized as an antiviral strategy in combination with current therapies to antagonize viral infectivity and propagation. Additionally, with information obtained from this study, small molecular modulators of miRNA function may be used to recapitulate miRNA antiviral function. Overall, our work highlights the role of miR-185 as a potential therapeutic miRNA that shows potent suppressive antiviral effects against SARS-CoV-2 infection. 

## 4. Methods

### 4.1. Reagents and Cell Culture

Huh7 human hepatoma cell lines were kindly gifted by Dr. Charles M. Rice (Rockefeller University, New York, NY, USA). Huh7 was cultured and maintained in Dulbecco’s Modified Eagle Medium (DMEM; Invitrogen, Carlsbad, CA, USA) supplemented with 10% fetal bovine serum (FBS; Gibco, Life Technologies, Waltham, MA, USA ) and 100 nM non-essential amino acids. HEK293T cells (CRL-3216) were purchased from ATCC and they were cultured and maintained in DMEM 10% FBS. Calu-3 cells were purchased from ATCC (HTB-55) and they were cultured and maintained in DMEM 10% FBS. pCMV14-3X-Flag-SARS-CoV-2 S was a gift from Zhaohui Qian (Addgene plasmid # 145780; http://n2t.net/addgene:145780; accessed on 2 August 2022, RRID:Addgene_145780) Accession# QHU36824. pCDNA3.3_CoV2_B.1.1.7 expressing SARS-CoV-2, UK strain (B.1.1.7) was a gift from David Nemazee (Addgene plasmid # 170451; http://n2t.net/addgene:170451; accessed on 2 August 2022 RRID:Addgene_170451). pcDNA3.3_CoV2_501V2 expressing SARS-CoV-2 Spike, 501V2 (B.1.351) was a gift from David Nemazee (Addgene plasmid # 170449; http://n2t.net/addgene:170449; accessed on 2 August 2022 RRID:Addgene_170449). pcDNA3.3-SARS2-B.1.617.1 expressing Spike of B.1.617.1 strain was a gift from David Nemazee (Addgene plasmid # 172319; http://n2t.net/addgene:172319; accessed on 2 August 2022; RRID:Addgene_172319). pcDNA3.3_SARS2_omicron_BA.1 expressing Spike protein from the omicron BA.1 variant was a gift from David Nemazee (Addgene plasmid # 180375; http://n2t.net/addgene:180375; accessed on 2 August 2022, RRID:Addgene_180375). SARS-CoV-2 isolate/Canada/ON/VIDO-01-2020 was used for SARS-CoV-2 infections. This virus was isolated from a patient at a Toronto hospital who had returned from Wuhan, China [58]. The second passage viral stock was sequenced (GISAID—EPI_ISL_425177) to confirm stability of the virus after culture in vDMEM on Vero-76 cells. All work with infectious SARS-CoV-2 viruses was performed in a Containment Level 3 (CL3) facility at the Vaccine and Infectious Disease Organization (VIDO) (Saskatoon, SK, Canada). HCoV-229E was a kind gift from Dr. Maxim Berezovski (University of Ottawa, ON, CA, USA) and was originally obtained from ATCC (VR-740). miR-185 mimics and inhibitors as well as negative control mimics and inhibitors (con-miR) were purchased from *mirVana* (Ambion, Austin, TX, USA). Lipofectamine RNAiMAX (Life Technologies, Carlsbad, CA, USA) was utilized to perform miRNA mimics and inhibitor transfections. Transfections were performed according to the manufacturer’s protocol with a ratio of 2.5 µL of RNAiMAX per 1 μL of 100 µM of miRNA mimic or inhibitors. Transfection of plasmid DNA was performed using Lipofectamine 2000 (Life Technologies, Carlsbad, CA, USA) as per manufacturer’s protocol. Fluvastatin was purchased from Sigma (SML0038) and 25-Hydroxycholesterol was purchased from Cayman Chemicals (11097).

### 4.2. Generation of D614G and N501Y Mutants

Site-directed mutagenesis was performed using a Quick-change Lightning kit (Agilent, Santa Clara, CA, USA) as per the manufacturer’s protocol with primers designed to introduce a point mutation which results in D614G and N501Y mutations in the pCMV14-3X-Flag-SARS-CoV-2 S plasmid. Primer sequences were as follows: for the N501Y mutation, Forward 5′ GGATTCCAGCCAACCTACGGCGTGGGTTACCAAC 3′ and Reverse 5′ GTTGGTAACCCACGCCGTAGGTTGGCTGGAATCC 3′; for the D614G mutation, Forward 5′ GTGCTGTACCAAGGCGTGAATTGCACAG 3′ and Reverse 5′ CTGTGCAATTCACGCCTTGGTACAGCAC 3′.

### 4.3. Production of Pseudotyped Viral Particles 

Hek293T cells in 10 cm culture dishes were transfected with 2 μg of plasmid-containing SARS-CoV-2 S glycoprotein (original Spike, Alpha, Beta or Delta variant) or mock plasmid and 4 ug HIV-1 NL4-3 ΔEnv Vpr Luciferase Reporter Vector (pNL4-3.Luc.R-E-). pNL4-3.Luc.R-E- was a kind gift from Dr Benoit Barbeau from University of Quebec in Montreal. After 72 h post-transfection, supernatants containing viral particles were harvested, centrifuged at 800× *g* for 5 min to remove cell debris and passed through 0.45 μm filter. 

### 4.4. Detection of S Protein of SARS-CoV-2 by Western Blot

Spike protein in cells or on pseudovirions were detected by Western blot using anti-SARS-CoV-2 S Monoclonal antibody (GeneTEX, GTX632604) predicted to recognize the Spike S2 subunit. Hek293T transfected with SARS-CoV-2-expressing plasmid and pNL4-3.Luc.R-E or mock was lysed using 1X SDS lysis buffer (50 mM Tris-HCl (pH 6.8), 2% SDS, and 10% glycerol). To pellet down pseudovirus, the viral supernatants were centrifuged at 27,800 rpm for 3 h in an AH-629 swinging bucket rotor at 4 °C through a 30% sucrose cushion, and virus pellets were resuspended in 100 μL SDS lysis buffer. Following lysis, the lysates were passed through a 21 G needle (BD Biosciences, San Diego, CA, USA). Lysates were separated on the 10% TGX stain-free gel (Bio-Rad, Hercules, CA, USA). The migrated proteins were visualized and activated on a ChemiDoc MP (Bio-Rad, Hercules, CA, USA). The proteins were then transferred onto a PVDF membranes using the Trans-Blot turbo (Bio-Rad, Hercules, CA, USA). Following the transfer, the membranes were blocked using 5% BSA in TBS-T. Subsequently, the membranes were incubated with primary antibodies overnight at various dilutions depending on the identity of the antibody. Following primary incubation, the membranes with secondary donkey anti-rabbit antibody were conjugated with horseradish peroxidase (1:20,000; Jackson ImmunoResearch Laboratories, Inc., West Grove, PA, USA) (115-035-062). The blots were visualized on the ChemiDoc MP (Bio-Rad, Hercules, CA, USA) with clarity ECL solution reagent (Bio-Rad, Hercules, CA, USA). Blot images were cropped and adjusted for contrast using Image Lab (Bio-Rad, Hercules, CA, USA).

### 4.5. Entry Assays 

Huh7 or A549^ACE2^ cells were seeded in 24-well plates and the next day the cells were transfected with either 100 nM con-miR, 50 nM con-inhibitor, 100 nM miR-185 mimic, or 50 nM inhibitor. Conversely, Calu-3 cells were reverse-transfected with mimics and inhibitors. On the following day, cells were transduced with 100 μl of media containing pseudovirions. Spinfection at 800× *g* for 1 h was performed to ensure efficient viral attachment. Cells were lysed 48 h post-transduction using 1X passive lysis buffer (Promega, Madison, WI, USA). Luciferase activity was measured using a microplate reader. All experiments were read in technical triplicates for at least three biological replicates.

### 4.6. Fluvastatin and 25-Hydroxycholesterol Treatments 

Huh7 cells were seeded in 24-well plates and the next day, cells were pre-treated for 24 h with varying concentrations of Fluvastatin, 25-HC or vehicle. Following pre-treatment, cells were transduced with equal amounts of pseudovirus. Spinfection at 800× *g* for 1 h was performed to ensure efficient viral attachment. Cells were lysed 48 h post-transduction using 1X passive lysis buffer (Promega, Madison, WI, USA). Luciferase activity was measured using a microplate reader. All experiments were read in technical triplicates for at least three biological replicates.

### 4.7. Transfections and Infections

Generally, for non-infected cells, transfections were conducted in 6-well plates. On day one, cells were reverse-transfected with con-miR or miR-185 mimics or inhibitors at a 100 nM final concentration for mimics or 50 nM final concentration for inhibitors. Transfections were conducted in OptiMEM using RNAiMAX reagent according to the manufacturer’s protocol. After 72 h post-transfection, cells were lysed for RNA analysis. For 229E infections, Huh7 cells or A549 Cells were used and infected with the HCoV-229E. Cells were initially seeded at 200,000 cells per well in a 6-well plate. The next day, the cells were forward-transfected, as previously described with con-miR or miR-185 mimic at a 100 nM final concentration. The following day, the cells were infected with HCoV-229E at an MOI of 0.05 in minimal serum-free media for 2 h. The medium was then replaced with complete medium following the 2 h incubation. Forty-eight hours post-infection, cells were lysed using Norgen total RNA isolation kit (Norgen Biotek, Thorold, ON, Canada) for total RNA isolation.

### 4.8. HCoV-229E Plaque Assay

Huh7 cells were seeded at 6.5 × 10^5^ cells/well in 6-well plates. The next day, the supernatants collected from HcoV-229E-infected cells were 10-fold serially diluted (1:10 to 1:10^6^). The media on Huh7 cells were replaced with 100 μL of serially diluted virus. Huh7 cells were incubated in virus-containing supernatant at 37 °C for 2 h. Following the 2 h infection, 2 mL of warm DMEM 1% Carboxymethylcellulose DMEM medium was added to each well and allowed to solidify. After 2–3 days, 2 mL 10% formaldehyde was added to each well and incubated for 30 min. Following formaldehyde fixation, media containing fixing solution were removed and staining solution (10% ethanol, 0.1% crystal violet) was added to each well and incubated for 15–30 min; excess stain was then washed with water and dried. Plaque forming units per milliliter of supernatant were calculated considering the dilution factor to report on the viral titers. 

### 4.9. Quantitative Real-Time PCR

For the RNA analysis, RNA isolations were performed using RNeasy Mini kit, (Qiagen, Mississauga, ON, Canada) or Norgen total RNA isolation kit (Norgen Biotek, Thorold, ON, Canada). For the RT-qPCR analysis, the isolated RNA was initially quantified using a NanoDrop (Thermo Fisher Scientific, Waltham, MA, USA). RNA integrity was evaluated using a 0.8% agarose RNA integrity gel in 1X Tris borate-EDTA (Ambion, Austin, TX, USA). In total, 500 ng of isolated RNA was reverse transcribed using the iScript reverse transcription kit (BioRad, Hercules, CA, USA) according to the manufacturer’s protocol. qPCR was subsequently performed using SSOAdvanced Universal SYBR GreenSupermix (Bio-Rad) according to the manufacturer’s instructions. A CFX Connect Real-Time PCR Detection System (Bio-Rad, Hercules, CA, USA) was utilized for the analysis of the qPCR runs. A final concentration of 500 nM of each evaluated primer was used in a final reaction volume of 10 μL. The 2^−ΔΔCt^ method was utilized to analyze the relative levels of mRNA and relative fold change in mRNA levels [59]. The 18sRNA was utilized to normalize samples. 

### 4.10. Transfections of miRNAs for SARS-CoV-2 Inoculation

Calu-3 cells (ATCC) were grown in complete DMEM prior to transfections. Cells were transfected in 6-well plates with miRNA-185 miRNA mimic or control miRNA (miRVana, ThermoFisher Scientific, Waltham, MA USA) via Lipofectamine RNAiMAX transfection reagent (Life Technologies, Waltham, MA, USA) in Optimem (Gibco, Life Technologies, Waltham, MA, USA) medium. The transfection reagents were added to the growth media of cells for a final concentration of 100 nM of miRNA. Calu-3 cells were also plated with no transfection reagent as a control. At 24 h post-transfection, media were removed from the cells. Cells were washed twice with 1X PBS before adding 100 μL of SARS-CoV-2/Canada/ON/VIDO-01/2020 at an MOI of 0.01. Supernatant and cellular RNA were collected for downstream analyses at 0, 24, 48 and 72 h post-inoculation. 

### 4.11. RNA Extraction from SARS-CoV-2 Infected Cells and Quantitative Real-Time PCR (qRT-PCR)

Cellular RNA was extracted using the RNeasy Mini kit (Qiagen, Hilden, Germany) according to the manufacturer’s instructions. vRNA was extracted supernatant using the QIAamp Viral RNA Mini Kit (Qiagen, Hilden, Germany). All cellular qRT-PCR was performed in triplicate on cDNA synthesized as previously described [60]. vRNA was quantified by Qiagen© Quanti-fast RT probe master mix (Qiagen, Hilden, Germany) using primer/probe sets specific for the SARS-CoV-2 subgenomic E gene. The reactions were performed on a StepOnePlusTM Real-Time PCR System in a 96-well plate (ThermoFisher Scientific, Waltham, MA, USA) as previously described [61].

## 5. Statistical Analysis

Data are presented as the mean of replicates, with error bars representing the standard error of the mean. Unless otherwise stated, statistical significance was evaluated using one-sample t-test and two-tailed Student’s t-test, and *p*-values less than 0.05 were deemed significant.

## Figures and Tables

**Figure 1 viruses-15-01921-f001:**
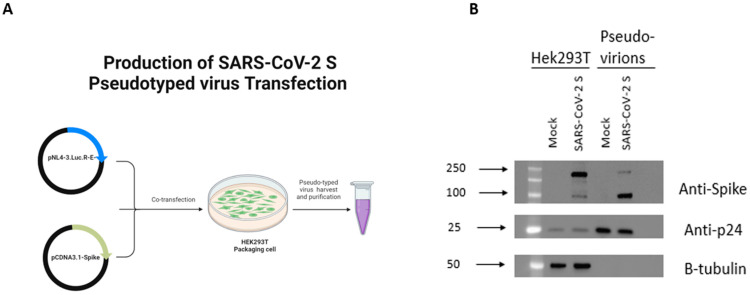
Detection of SARS-CoV-2 S protein in Hek293T cell lysates and on pseudotyped virus. (**A**) Schematic representation of methodology utilized to produce Spike protein pseudotyped virus. (**B**) Western blot analysis of Spike protein in HEK293T lysates used for virus production and in virus particles. Anti-Spike was used to verify expression of Spike protein in cell lysates and pseudotyped virus. HIV anti-p24 and B-tubulin antibody were used as loading control for virions and cell lysates, respectively. Figure 1A generated using Biorender.

**Figure 2 viruses-15-01921-f002:**
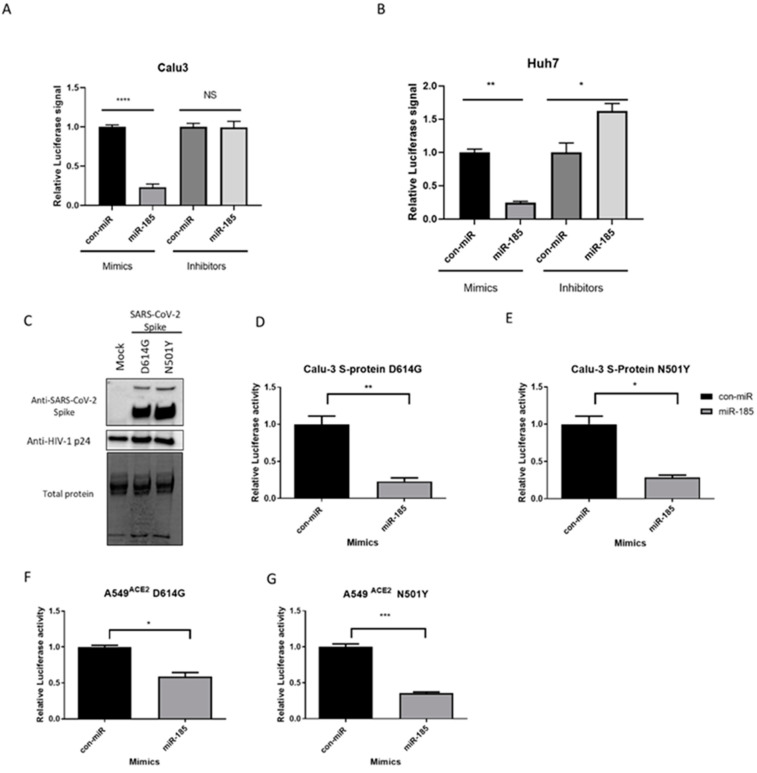
miR-185 inhibits pseudovirions entry in Huh7 hepatoma cell line and Calu-3 lung carcinoma cell line. (**A**) Huh7 cells were forward-transfected with miR-185 mimics/inhibitors or con-miR mimics/inhibitors and (**B**) Calu-3 cells were reverse-transfected with miR-185 mimics/inhibitors or con-miR mimics/inhibitors. After 24 h post-transfection, the cells were infected with SARS-CoV-2 S-pseudotyped virus. (**C**) Western blot analysis of Spike protein on pseudotyped virus particles. Anti-Spike was used to verify expression of Spike pseudo-typed virus and HIV anti-p24 for virions was used as loading control for virions. (**D**) Calu-3 cells infected with S-protein D614G pseudotyped virus (**E**) Calu-3 cells infected with S-protein N501Ypseudotyped virus (**F**) A549^ACE2^ cells infected with S-protein D614G pseudotyped virus (**G**) A549 ^ACE2^ cells infected with S-protein N501Y pseudotyped virus. After 48 h post-infection, cells were lysed in 1X passive lysis buffer and Luciferase activity was measured using a microplate reader. Error bars represent the means ± SEM of at least three biological replicates. * *p* < 0.05, ** *p* < 0.01, *** *p* < 0.001, **** *p* < 0.0001, NS not significant.

**Figure 3 viruses-15-01921-f003:**
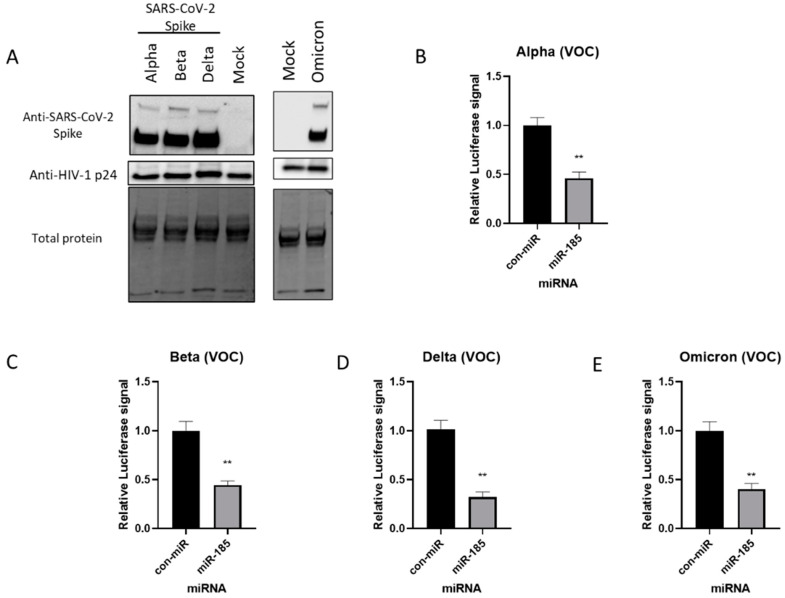
miR-185 inhibits entry of SARS-CoV-2 Alpha-, Beta-, and Delta-Spike-variant-pseudotyped virus in cell culture. (**A**) Western blot analysis of Spike protein on pseudotyped virus particles. Anti-Spike was used to verify expression of Spike pseudotyped virus and HIV anti-p24 for virions was used as loading control for virions. Calu-3 cells were transfected with miR-185 mimics or con-miR mimics and 24 h post-transfection, the cells were infected with media containing SARS-CoV-2 S-pseudotyped virus (**B**) Alpha (**C**) Beta and (**D**) Delta variants of concern (**E**) Omicron variant of concern. After 48 h post-infection, cells were lysed in 1X passive lysis buffer and Luciferase activity was measured using a microplate reader. All experiments were read in technical triplicates for at least three biological replicates. Error bars represent the means ± SEM of at least three biological replicates. ** *p* < 0.01.

**Figure 4 viruses-15-01921-f004:**
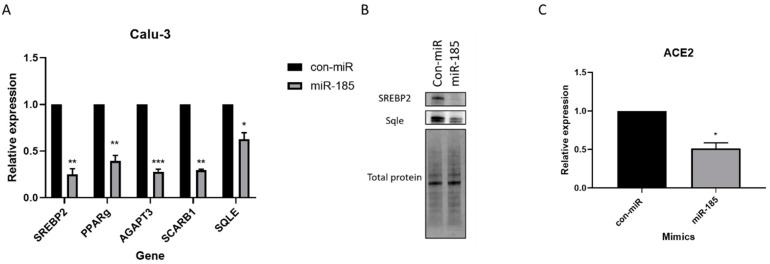
miR-185 inhibits entry by modulating lipid metabolism and repression of ACE2 expression in Calu-3 cells. Calu-3 cells were reverse-transfected with miR-185 mimics/inhibitors or con-miR mimics/inhibitors. After 72 h post-transfection, cells were lysed for RNA analysis. (**A**) qRT-PCR analysis of lipogenic genes in con-miR and miR-185-transfected cells. (**B**) Western blot analysis of SREBP2 and SREBP2 target, SQLE (**C**) ACE2 levels in con-miR and miR-185-transfected cells. Error bars represent the means ± SEM of at least three biological replicates. * *p* < 0.05, ** *p* < 0.01, *** *p* < 0.001.

**Figure 5 viruses-15-01921-f005:**
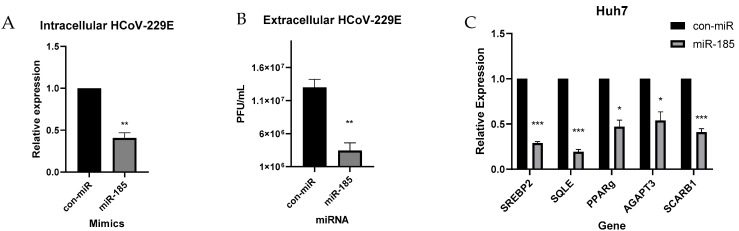
miR-185 inhibits HCoV-229E replication and infectivity. Huh7 cells were transfected with miR-185 mimics or con-miR mimics. After 24 h post-transfection, cells were infected with HCoV-229E at an MOI of 0.05. Cells lysed for RNA analysis of intracellular levels of virus 48 h post-infection. (**A**) Plaque assay of extracellular levels of virus. (**B**) Relative levels of extracellular virus from con-miR and miR-185-transfected cells. (**C)** qRT-PCR analysis of lipogenic genes in con-miR and miR-185-transfected cells. Error bars represent the means ± SEM of at least three biological replicates. * *p* < 0.05, ** *p* < 0.01, *** *p* < 0.001.

**Figure 6 viruses-15-01921-f006:**
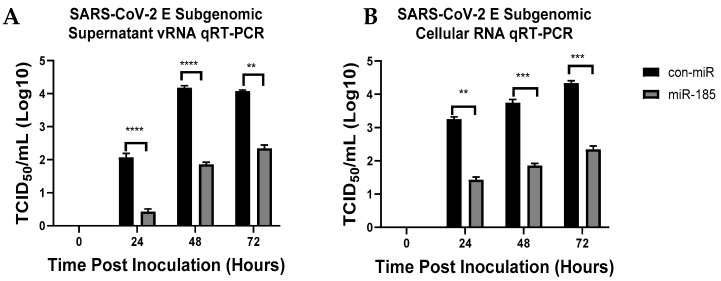
miR-185 results in significant inhibition in SARS-CoV-2 intracellular and extracellular levels at various time points post-inoculation. (**A**) Quantification of extracellular SARS-CoV-2 subgenomic levels from Calu-3 transfected with either con-miR or miR-185 at 24 h, 48 h and 72 h post-inoculation. (**B**) Quantification of intracellular SARS-CoV-2 subgenomic levels from Calu-3 transfected with either con-miR or miR-185 at 24 h, 48 h and 72 h post-inoculation. Error bars represent the means ± SEM of at least three biological replicates. Two-way ANOVA was used to determine the statistical significance. ** *p* < 0.01, *** *p* < 0.001, **** *p* < 0.0001.

## Data Availability

The data presented in this study is available upon reasonable request from the Corresponding author.

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
