# Peer review of "microRNA-185 Inhibits SARS-CoV-2 Infection through the Modulation of the Host’s Lipid Microenvironment"

_viruses, 2023, doi:10.3390/v15091921_

Round 1
Reviewer 1 Report
This research article investigate the role of miR-185 in modulating the entry of SARS-CoV-2 using different models (i) a spike protein pseudotyped virus with or without specific mutations (D614G and N501Y) and clades, (ii) in low pathogenicity coronavirus 229E infection (HCoV-62 229E) and replication-competent SARS-CoV-2.
The authors have previously shown that miR-185 54 inhibits hepatitis C and dengue viruses through alterations in the host’s lipid microenvironment and others study’s have demonstrated the impact of depleting membrane cholesterol on infection of SARS-CoV-2 and other coronaviruses. The authors clervely explored the potential role of miR-185 on SARS-CoV-2 infection and its lipid environment.
The subject is very interesting but deserves to be better framed by additional controls thus ensuring the quality of the results.
Major remarks:
- Ensure the correct conformation of the spike proteins expressed on the surface of the pseudoparticles by carrying out a neutralization test.
- Ensure cell viability after transfection
- Provide control of miR transfection with a miR-185 carrying a fluorophore as control
- Present the results according to the RLU values ​​detected because the presentation in the form of relative luciferase signal does not make it possible to assess the level of infection and the differences between the different tests on the different cell lines.
- Specify the target (which spike subunit) of the anti-spike antibody used?
- Specify the clade and if available the accession number of the sequences used for the different variants of SARS-CoV-2
- Indicate at which MOI, are the cells infected with the pseudoparticles?
- Figure 5B: the titers obtained 48 hours after infection with the HcoV-229E strain in PFU/ml are very high following an MOI of 0.05. The scale is not correct from 0 to 1.5x 107.
Minor remarks:
- P3 – line 105: viral instead of vial
- Fig 2 : Calu-3 instead of Calu3
- P4 – line 116: repetition of with
- P5 – line 147: choose between SARS-CoV-2 or Sars-CoV-2
- P11 – line 431: 25-Hydroxycholesterol instead of 25-Hydroxychlesterol
- Fig 2S: Fluvastatin instead of Fluvastain
- Fig 3S: Contradiction on the indications of the cell lines used between the title of the figure and the legend.
Author Response
Reviewer 1:
- Ensure the correct conformation of the spike proteins expressed on the surface of the pseudoparticles by carrying out a neutralization test.
Response: We thank the reviewer for their comment. We do agree with the reviewer that ensuring the functionality and conformation of the spike protein is essential. Thus, to confirm this we have performed an additional experiment using a mock virus which does not express the spike protein on its surface (Figure S1). When we use this virus to infect cells, a low RLU is observed relative to the spike expressing virus, validating that the entry signal observed in our assay is spike-based and thus the confirmation is not compromised. Although it would be interesting to further confirm the confirmation of the Spike using a neutralization assay, we believe it is not necessary since the entry assay adapted in our paper is well validated and widely used to assess entry of pseudotyped viruses- and has been found to accurately report on spike-based entry as described in the following papers (Hoffmann et al., 2020) DOI: 10.1016/j.cell.2020.02.052 and (Capcha et al., 2021). Additionally, the focus of this paper is to highlight the effects of miR-185 miRNA on the entry, which is clearly demonstrated in our experiments and validated to be a specific response using our experiment presented in figure S1.
- Ensure cell viability after transfection
Response: We have performed a cell viability assay (MTT assay) which has confirmed that miRNA transfection does not affect cellular viability at various time points. Figure S6.
- Provide control of miR transfection with a miR-185 carrying a fluorophore as control
Response: We thank the reviewer for their comment. We have validated the transfection efficiency by evaluating the knockdown of direct targets of the miRNA, which is a common method for evaluation and confirming the transfection (Figure 4A and 4B) and Figure S8. This not only confirms the transfection of the miRNA but also validates the functionality of the transfected mimics.
- Present the results according to the RLU values ​​detected because the presentation in the form of relative luciferase signal does not make it possible to assess the level of infection and the differences between the different tests on the different cell lines.
Response: We thank the reviewer for their comment. The reason behind the use of Relative Luciferase values (normalized RLU) as opposed to RLU in our study is to normalize the signal and take into account the interassay variability and differences between virus batches. This results in emphasizing the effects of miR-185 on entry which is the focus of the study. However, we do agree that additionally providing the RLU data would complement the data set provided and allow to compare the levels of infection between different experiments. This data was added to the manuscript (Figures S2 qnd S3).
- Specify the target (which spike subunit) of the anti-spike antibody used?
Response: Based on sequence analysis, this antibody is predicted to recognize S2 subunit. This antibody was validated to be able to detect multiple SARS-CoV-2 VOCs, including Omicron variant.
- Specify the clade and if available the accession number of the sequences used for the different variants of SARS-CoV-2.
Response: We have updated our methods sections to include more information about the sources of the plasmids and the variants used in the study; available accessions numbers were added.
- Indicate at which MOI, are the cells infected with the pseudoparticles?
Response: We have updated our manuscript methods section to include the MOI used for our infections.
- Figure 5B: the titers obtained 48 hours after infection with the HcoV-229E strain in PFU/ml are very high following an MOI of 0.05. The scale is not correct from 0 to 1.5x 107.
Response: Huh7 is a highly permissive cell line for HcoV-229E infection, thus it is not surprising that higher viral titres are observed. Other studies have reported similar titers after the infection of Huh7; some of these studies include (de Wilde et al, 2018) https://doi.org/10.1016/j.virol.2017.11.022 and (Shaban et al., 2021) DOI: 10.1038/s41467-021-25551-1. Despite the titer levels, a relative decrease in viral titers in observed in miR-185 transfected cells compared to con-miR transfected cells, confirming the inhibitory effects of miR-185 on the pathological of the virus. This data is complemented with the inhibitory effects of the miRNA on SARS-CoV-2 as seen in figure 6.
We have adjusted the scale in figure 5B.
Minor remarks:
- P3 – line 105: viral instead of vial
Adjusted
- Fig 2 : Calu-3 instead of Calu3
Adjusted
- P4 – line 116: repetition of with
Adjusted
- P5 – line 147: choose between SARS-CoV-2 or Sars-CoV-2
Adjusted
- P11 – line 431: 25-Hydroxycholesterol instead of 25-Hydroxychlesterol
Adjusted
- Fig 2S: Fluvastatin instead of Fluvastain
Adjusted
- Fig 3S: Contradiction on the indications of the cell lines used between the title of the figure and the legend.
Adjusted

Reviewer 2 Report
The authors tried to analyze the effects of miR-185 on the modulation of SARS-CoV-2 entry and replication. The authors found that miR-185 affected the entry and the infectivity of SARS-CoV-2, using pseudotyped virus, surrogate virus, and live virus. The expression of lipid metabolism pathways-associated genes and ACE2 gene was reduced by miR-185. These results suggest new future antiviral strategies.
The explanation was not described why different cells were used in different experimental assays. Calu-3 cells were used in Figure 3, 4, and 6. However, Huh7 cells were used in Figure 5.
In Figure 6A and 6B, “Subegenomic” would be “Subgenomic”.
Minor editing of English language required.
Author Response
Reviewer 2:
The explanation was not described why different cells were used in different experimental assays. Calu-3 cells were used in Figure 3, 4, and 6. However, Huh7 cells were used in Figure 5.
Response: We thank the reviewer for their comment, the reason behind using Huh7 in some experiments and Calu3 in the others is related to the permissiveness of the virus used. For example, Huh7 is an optimal host for HCoV-229E virus, however Calu-3 isn’t due to its lack of expression of the viral receptor. We have now updated our manuscript to clarify this.
In Figure 6A and 6B, “Subegenomic” would be “Subgenomic”.
Adjusted
